# The Effect of 9-Week Dietary Intervention on Anthropometric Parameters and Blood Pressure in Children with Excessive Body Weight

**DOI:** 10.3390/metabo15090621

**Published:** 2025-09-18

**Authors:** Karolina Gajda, Marta Jeruszka-Bielak, Magdalena Górnicka, Irena Keser, Jadwiga Hamulka

**Affiliations:** 1Department of Human Nutrition, Institute of Human Nutrition Sciences, Warsaw University of Life Sciences (SGGW), 02-787 Warsaw, Poland; karolinagajda@interia.pl (K.G.); magdalena_gornicka@sggw.edu.pl (M.G.); 2Laboratory for Nutrition Science, Faculty of Food Technology and Biotechnology, University of Zagreb, 10000 Zagreb, Croatia; irena.keser@pbf.unizg.hr

**Keywords:** children, overweight and obesity, dietary intervention, body mass reduction program, anthropometric parameters, blood pressure

## Abstract

Background/Objectives: The prevalence of pediatric overweight and obesity is steadily increasing, posing a tremendous problem and challenge to public health. Various strategies have been undertaken to combat this issue with mixed results. This study aimed at assessing the effect of a 9-week dietary intervention on anthropometric measurements and blood pressure among children aged 7–12 years with excessive body weight. Methods: The main aim of the intervention was to improve the children’s eating habits according to dietary guidelines for Polish children and adolescents with an individual approach and dietitian support. The intervention was completed by 68 children, 38 girls and 30 boys. Data on nutrition; anthropometrics, including body composition; and blood pressure were collected at baseline and after 3, 6, and 9 weeks of the dietary intervention. Results: The 9-week dietary intervention resulted in significant reductions in BMI z-score, fat mass, and body weight of 10.7%, 7.6%, and 4.0%, respectively. Blood pressure, especially diastolic blood pressure, also significantly decreased. Moreover, positive changes in dietary behaviors were noted. Conclusions: A 9-week dietary intervention based on an appropriate, high-quality diet in line with the Healthy Eating Pyramid guidelines, with individual counselling and constant monitoring of the implemented changes, an individual approach, support from a dietitian, and the involvement of parents or guardians, can be an effective tool for starting to improve the diets and health of overweight children.

## 1. Introduction

The prevalence of overweight and obesity in child populations is steadily increasing in high-, middle-, and low-income countries, posing a tremendous problem and challenge to public health [1,2]. According to the World Health Organization (WHO), the percentage of children and adolescents aged 5–19 years with excessive body weight has risen dramatically from just 8% in 1990 to 20% in 2022, similarly among both boys and girls [3]. Data from the WHO European Childhood Obesity Surveillance Initiative (COSI), round 6 (2022–2024), indicate that as much as 25% of children aged 7–9 years in Europe have excessive body weight, with a higher prevalence among boys (26%) compared to girls (23%) [4]. Similar results were obtained for adolescents within the Health Behaviour in School-aged Children (HBSC) study, which showed that 22% of adolescents had overweight or obesity, although greater differences between boys (27%) and girls (17%) were found [5]. The problem is even more pronounced in Poland, as the prevalence of excessive body weight among Polish children is 33% and is one of the highest in Europe and the highest in Eastern Europe [4]. Furthermore, Poland has one of the highest projected childhood obesity growth rates for the period 2020–2035, at 4.8% per 5 years [6]. This rate is much higher than for most European countries and even for the USA. For comparison, it is approximately 2.5% for Germany, Greece, Spain, and France; approximately 3.0% for the UK, Austria, Finland, and Norway; and 2.4% for the USA.

Obesity has been recognized as the most common metabolic disease and is characterized by increased body weight due to increased fat mass caused by adipocyte hypertrophy and/or hyperplasia, the last typically occurring during childhood or adolescence [7,8]. It is not only perceived as a health problem per se, but it also contributes substantially to the development of numerous non-communicable diseases, including type 2 diabetes, cardiovascular diseases, and certain cancers [7]. Additionally, it brings economic, psychological, and social consequences, as well as decreases in the quality and length of life [8]. Children and adolescents with excessive body weight are at increased risk of mental and emotional disorders, such as depression, eating disorders (e.g., compulsive overeating disorder), anxiety disorders, and lower self-esteem, as well as problems with stigmatization and a sense of discrimination [9,10]. Additionally, obesity established in childhood is likely to persist throughout life and may contribute to the above-mentioned problems and premature mortality [11]. Therefore, from a public health perspective, preventive strategies and early-introduced obesity management which focus on modifiable risk factors, starting from an early age, are extremely important.

Recent research findings shed new light on the pathogenesis of obesity and the potential for its prevention and treatment. Biological predisposition (e.g., genetic variants), socioeconomic forces, and environmental factors together promote the development of obesity and resistance to efforts at obesity management [12,13,14].

Obesity management needs the introduction of interventions, including dietary interventions. These might consist of nutrition education alone or such education might be combined with moderate energy restriction and structured dietary plans [12]. However, inconsistency in the definition of moderate energy reduction according to pediatric age and disease severity is observed [15]. Improving baseline diets and the selection of dietary strategies that are informed by individual preferences and circumstances, family environments, and available support may be appropriate [12]. Recently, a position statement on medical nutrition therapy in the management of overweight or obesity in children and adolescents, prepared by an expert committee convened by the European Association for the Study of Obesity (EASO) and developed in collaboration with the European Federation of the Associations of Dietitians (EFAD), has been published [14]. It includes seven recommendations, including delivering long-term and regular individual treatment by a dietitian that can result in the maintenance of energy deficits that reduce adiposity indicators in children and adolescents with obesity, while maintaining nutritional requirements for growth; focusing on reduced energy density through increased vegetable consumption, adequate fruit intake, and limited fruit juice consumption; introducing simple but explicit food-based guidance towards achieving country-specific guidelines, which should be delivered as part of a behavior change strategy; and parental involvement that is age-appropriate, transitioning from parent-focused involvement for younger children to adolescent-focused involvement for children aged 12 years and older.

Although numerous interventions have been conducted in pediatric populations, the results remain inconclusive. Moreover, they differ in many aspects, including the age and the number of participants, the scope of the intervention and its duration, and the outcomes used for the evaluation of their effectiveness [15].

Thus, this study aimed to assess the effect of a 9-week dietary intervention on anthropometric measurements and blood pressure among children aged 7–12 years with excessive body weight. A focus on diet quality and not nutrient quantity, an individualized approach, and high involvement of parents/guardians, as well as regular support from a dietitian, were the main features of the intervention.

## 2. Materials and Methods

### 2.1. Study Design and Participants

The study was approved by the Ethics Committee of the Food and Nutrition Institute (Resolution No. 2606/2012). The guidelines of the Declaration of Helsinki were followed during the study, and written informed consent was obtained from parents/caregivers for their children’s participation.

The study was conducted among children whose parents responded to program advertisements placed on the Internet, in medical centers, and in primary schools. In total, 250 children were put forward for the program, but due to the inclusion and exclusion criteria, only 111 children qualified for the study (Figure 1). The study eligibility criteria were as follows: child’s age (7–12 years); good general health; lack of metabolic disorders; amenorrhea in girls; written consent of parents/guardians to participate in the study. The exclusion criteria included the following: child’s age <7 or >12 years; low or normal body weight; previous treatment for obesity; secondary obesity; metabolic diseases (hypothyroidism/hyperthyroidism, type 1 diabetes, or adrenal gland diseases); pharmacotherapy; mental disorders; implanted pacemakers or implants. Excessive body weight was diagnosed during a visit to a general practitioner based on the BMI centile charts developed within the OLAF project and interpretation according to the International Obesity Task Force (IOTF) criteria [16,17]. Overweight was diagnosed when the BMI centile was within the range of 85–<95, while for obesity a BMI centile ≥ 95 was used.

After the initial selection, 97 children, including 52 girls and 45 boys, reported for the baseline visit and started the body weight reduction program. The 9-week dietary intervention was completed by 68 children (61% of the entire group qualified for the study), 38 girls and 30 boys. Withdrawal from the program mainly resulted from the perceived difficulties in engaging in the program (*n* = 22), the necessity of participating in three follow-up visits every 3 weeks (*n* = 8), illness (*n* = 7), or decreased motivation (*n* = 16). No significant differences were found between completers and non-completers in baseline characteristics.

All visits took place at the anthropometric laboratory in the Department of Human Nutrition at the Warsaw University of Life Sciences. At each visit, a child together with a parent/guardian participated in the meeting. Prior to the study, all participants were informed about the purpose and assumptions of the body weight reduction program and about the possibility of withdrawing from participation at any stage of the program without the need to justify the decision. The parent/guardian signed an informed consent form allowing their child to participate in the study and received instructions on the child’s preparation for anthropometric measurements and on the completion of the 3-day food record. Additionally, parents filled in a questionnaire that aimed to determine dietary habits, assessing selected lifestyle and socio-demographic factors.

During the baseline and follow-up visits, a child’s anthropometric and blood pressure measurements were taken to monitor the changes in those parameters. Dietary data were collected with the use of a 3-day food record prior to all visits. At each follow-up visit the achieved results were discussed, everyday problems related to the implementation of nutritional recommendations were solved, and the children and their parents/guardians were motivated to continue the program.

### 2.2. Dietary Intervention

The dietary intervention was based on a 9-week body weight reduction program that aimed at a gradual and slight body weight reduction of approx. 0.5 kg per week. The main aim of the intervention was to improve the children’s eating habits successively and gradually. Energy requirements were calculated for each participant with the formulas for total energy expenditure [18], and ideal body weights were calculated according to the body mass index method [19] and the BMI centile charts for the Polish population [16]. The goals for macronutrients were adjusted to the reference intake ranges for children and adolescents [18] and equaled 15–20% of energy for proteins, ≤30% of energy for total fats (including <10% for saturated fatty acids), and 45–65% of energy for carbohydrates (including <10% for sugars). An individual dietary model was applied to reflect the current state of a child’s dietary behaviors. Appropriate selection of products that are good sources of dietary fiber and products with a low glycemic index (GI) was recommended [20]. It was also proposed to change the eating habits of the whole family to support a child in introducing the changes in his/her dietary behaviors.

Recommendations reflected the “Healthy Food Pyramid” and the 10 Principles of Healthy Eating for Children and Adolescents developed by the Institute of Food and Nutrition in Warsaw [21]. Children were instructed to eat regularly 5 meals a day, including 3 main meals and 2 snacks, every 3 h during the day. Additional snacking between meals, found to be typical for the majority of this group, was not recommended to eliminate the habit of snacking and to maintain a stable level of glycemia and insulin secretion between meals. It was emphasized to eat breakfast before going to school and to eat the last meal at least 2 h before bedtime.

Generally, dietary recommendations included the following:Limited purchasing and consumption of sugar, sweets, candied fruit, sweetened beverages, fruit juices, high-fat products, fast food, and savory snacks;Increased access to and consumption of products such as vegetables, especially raw ones, and fruit, with special attention paid to GI;Introducing whole-grain products like bread, oats, groats, and whole-grain pasta to reduce the feeling of hunger;Drinking the appropriate amount of fluids, mainly water;Eating daily at least three portions of protein food sources like low-fat dairy, eggs, pulses, lean meats, and fatty fish;Greater variety in diet;Using appropriate culinary technics for meal/dish preparation, avoiding frying with fat;Mindful eating without any disruptors like phones, TV, etc.

### 2.3. Nutrition Education

The dietary intervention included four sessions of individual counselling of a child and her/his parents or guardians with a dietitian, during which nutrition education and correction of dietary errors were provided. Dietary rules and guidelines, portion sizes, and recommended culinary techniques were presented and discussed. Some practical tasks were introduced to verify the children’s nutrition awareness and knowledge. Participants received written materials, i.e., dietary recommendations and a 7-day meal plan model for proper meal composition individually adjusted to their needs and goals; however, they were informed that they could exchange products within the respective food groups. They also received lists of recommended and contraindicated food products, as well as products with a low glycemic index. Each child was given a “diary” and asked to note the foods and beverages (without detailed data and quantities) that were consumed daily, which enabled tracking of his/her adherence to the dietary plans between visits.

Parents could and did contact the dietitian by phone or e-mail at any time during the study.

During the body weight reduction program, previous levels of physical activity were maintained for the verification of the role of the dietary intervention in regulating body weight. Children participating in the study did not introduce any additional forms of physical activity during the study.

### 2.4. Dietary Assessment

The 3-day food record method was used to collect data on eating habits at baseline and after 3, 6, and 9 weeks of the dietary intervention. Children, together with their parents/guardians, filled in the questionnaire and noted all types and amounts (using household measures) of foods and drinks consumed during 3 non-consecutive days, 2 weekdays, and 1 weekend day. Data were checked and verified by the dietitian during each visit, and additional information was obtained if needed. Data from the baseline to the 9-week visit were entered into the Dieta 6.0 computer program, and the intake of energy and selected nutrients was calculated and referred to the Polish nutritional recommendations according to age, gender, and physical activity level (for energy) [18]. Data from the 3-day food record completed prior to the visits after 3 and 6 weeks were used to verify the implementation of nutritional goals during the intervention.

Furthermore, we also calculated the Diet Quality Index (DQI) that assessed the quality of a child’s diet at baseline and after 9 weeks of intervention. It consisted of eight variables which reflected compliance with the nutrition recommendations and dietary guidelines, namely, the intake of total fat, saturated fatty acids, cholesterol, protein, sodium, and calcium, as well as the consumption of vegetables and fruits, breads, and cereals (Table 1). Each element was scored 0, 1, or 2 points, and all points were summed. A lower score meant better diet quality [22]. According to the DQI, dietary quality was classified as follows: good (0–4 points in total), sufficient (5–12 points), or poor (13–16 points).

### 2.5. Anthropometric and Blood Pressure Measurements

During the body weight reduction program, anthropometric and blood pressure measurements were performed four times by one dietitian with the usage of the same instruments.

Body height (in cm) was measured with the HR-001 stadiometer produced by Tanita in the straight position, with the head in the horizontal Frankfort plane. Body weight measurement (in kg) was conducted with the BC 420 scale produced by Tanita (CE certificate, meeting the requirements of the MDD 93/42EEC directive; Tokyo, Japan). A child was measured in their underwear, without shoes, in a standing position, with their weight distributed evenly on both legs, the upper limbs hanging freely along the body. Additionally, waist (WC, in cm) and hip (HC, in cm) circumferences were measured with a stretch-resistant tape that provides a constant 100 g tension (SECA 201, Hamburg, Germany). Measurement of WC was taken at the point midway between the iliac crest and the costal margin (lower rib) on the anterior axillary line in a resting expiratory position, while for HC, measurements were taken at the widest part of the buttocks, with the tape parallel to the floor.

All measurements followed the International Standards for Anthropometric Assessment [ISAK] recommendations [23].

Body mass index (BMI) was calculated, and the results were interpreted as described above. Moreover, the BMI z-score was calculated using the WHO centile charts according to the following formula:BMI z-score = [actual child’s BMI − BMI norm (50th c.)]/standard deviation for BMI norm

The WHO recommends a BMI z-score ≥ 1 as the cut-off for overweight and a BMI z-score ≥ 2 as the cut-off for obesity [24].

Additionally, Cole’s index (LMS, Least Mean Square), enabling a percentage assessment of BMI in relation to the standard BMI corresponding to the 50th percentile, was calculated with the following formula:LMS = [actual child’s BMI x100%]/BMI norm (50th c.)

LMS > 120% was interpreted as obesity, while for overweight the range of 110–119% was used [17].

The waist-to-height ratio (WHtR) index with a threshold value ≥ 0.50 was used for assessing abdominal obesity [25].

Body composition was measured with the usage of the non-invasive bioelectrical impedance analysis (BIA) method and the Maltron International BioScan 920-2 device (Rayleigh, UK). The test was performed according to the manufacturer’s instructions. The analysis was performed in an eight-electrode system in the contralateral configuration, in a supine, relaxed position, with the limbs positioned at an angle of 30–45° to the body. Disposable electrodes were placed on the dorsal surface of the upper limbs (above the wrist joint gap) and lower limbs (ankle joint) and 3 cm below the above structures. The site of electrode application was previously washed with alcohol to degrease the body and remove contamination. The children were asked to refrain from drinking any liquids and eating any foods for 4 h before and to avoid intense physical exercise for 12 h before the analysis. The body composition analysis was always performed under the same conditions, i.e., after a 10 min rest, without shoes or socks, after emptying the bladder. The following components of body composition were analyzed: fat mass (FM); fat-free mass, including muscle mass; and total body water, extracellular water, and intracellular water. For the purpose of this paper, only FM was used, as well as the fat mass index (FMI), which was calculated with the following formula:FMI = FM (kg)/height^2^ (m^2^) 

Blood pressure was measured with an A&D Medical UA-631 automatic upper-arm blood pressure monitor (Tokyo, Japan), using a pediatric cuff. Measurements were taken in accordance with the principles adopted in the 4th Report of the National High Blood Pressure Program Working Group on High Blood Pressure in Children and Adolescents [26] in identical conditions. Children sat with their backs resting on the back of a chair, with the soles of their feet on the ground, with their right arm supported so that the elbow was at the level of the heart, after at least 10 min of rest. The cuff was placed on the right arm at the level of the heart. Blood pressure was measured three times at 1–2 min intervals. The arithmetic mean of the three measurements was used for analysis. The results for both systolic (SBP) and diastolic (DBP) blood pressure were compared with the reference values based on the OLAF project centile charts for the Polish population. A result between the 5th and 95th centile was considered to be normal [27].

### 2.6. Statistical Analysis

The statistical software package IBM SPSS Statistics 24.0 (IBM, 2016, New York, NY, USA) was used to conduct the analyses. The results are presented as means, standard deviations, medians, and ranges (minima–maxima) for continuous variables and as percentages for categorical variables. The assumption of a normal distribution for continuous variables was verified with the Kolmogorov–Smirnov test. In order to determine the statistically significant differences among subsequent measurements, we used the Wilcoxon test for quantitative dietary assessment or McNemar’s test for qualitative dietary assessment when two time-points were compared. For anthropometric and blood pressure measurements, when four time-points were compared, the Friedman test was applied. To analyze the statistical differences between girls and boys, the Mann–Whitney U test and the Pearson chi-square test were used for continuous and categorical variables, respectively. The threshold for statistical significance was set at *p* ≤ 0.05.

## 3. Results

### 3.1. Characteristics of Particpants

Participant characteristics are presented in Table 2 for the whole group and stratified by sex. The study included 68 children, with a slightly higher proportion of girls (56%), children belonging to the older (10–12 years) age group (59%), and town inhabitants (53%). The majority of parents, especially mothers, were aged 19–30 years. Among mothers, secondary education dominated (42%), while among fathers, vocational education was the most frequent (43%). Assessing family obesity, it was noted that it occurred more often in fathers (18.5%), with more obese mothers and fathers found in the group of girls. There were no statistically significant differences between girls and boys in their socio-demographic characteristics or in the occurrence of excess body weight in their parents.

### 3.2. Dietary Changes as Effects of the Intervention

During the dietary intervention, significant and positive changes were demonstrated in energy and selected nutrient intakes, with the exception of protein (Table 3). There were no significant differences between boys and girls; hence, the results are presented for the total population. The intake of energy, fat (as % of total energy), and saccharose decreased by 16%, 8%, and 13%, respectively. By contrast, the intake of dietary fiber and selected minerals and vitamins increased by at least 10% after 9 weeks of the dietary intervention.

As a result of the dietary intervention, dietary quality also improved in the total population and in both gender subgroups. At baseline, none of the menus were evaluated as good according to the DQI, while after the 9-week intervention the diet quality of eight children (four girls and four boys) was classified as good (Table 4). Moreover, none of the diets was assessed as poor after the intervention.

### 3.3. Anthropometric and Fat Mass Changes as Effects of the Intervention

As a result of the body weight reduction program, the values of somatic indexes, i.e., BMI z-score, LMS, WC, and WHtR, decreased significantly (Table 5), similarly to fat mass content (FM and FMI; Table 6). Although the changes were found in the total population and in both genders, higher percentage changes were shown among girls than boys, especially for BMI z-score (14.3% vs. 3.0%), LMS (4.9% vs. 1.5%), FM (9.1% vs. 5.7%), and FMI (9.2% vs. 5.3%). Generally, the greatest improvement in BMI z-score, LMS, WC, and WHtR was noted after 3 and 9 weeks, while for FM and FMI it was noted after 3 weeks, with only slight or a lack of further changes at the end of the dietary intervention. The intervention also significantly contributed to a reduction in body weight by an average of 2.2 kg over a 9-week period, i.e., 0.2 kg per week, which was slightly lower than the recommended value of 0.5 kg per week.

### 3.4. Blood Pressure Changes as Effects of the Intervention

The dietary intervention also had a positive effect on the normalization of blood pressure in the population (Table 7). In all children, systolic and diastolic blood pressure values were higher at the beginning than after completion of the program, with average decreases of 3.6% and 5.4% for SBP and DBP, respectively. Before the intervention, the average values for SBP and DBP were close to the 95th percentile, the threshold for a diagnosis of elevated blood pressure in children [27]. Moreover, too high blood pressure was found in 51.5% of the children. For both SBP and DBP, the greatest decrease was noted after 6 weeks of the dietary intervention, and better normalization occurred in the case of DBP.

## 4. Discussion

This study examines the effect of a dietary intervention in managing excessive body weight in children aged 7–12 years. It consisted of a 9-week body weight reduction program with individual family-dietitian counselling and a major goal of improving the children’s eating habits according to dietary guidelines.

After the 9-week dietary intervention, significant reductions in body weight, BMI z-score, Cole’s index, waist circumference, and WHtR, as well as fat mass and fat mass index, were found. Although the differences were significant regardless of gender, they were more pronounced in girls than in boys, and the reduction was in the range of 3.4–14.3% for girls and 1.5–5.7% for boys, depending on the variable. Moreover, the decrease in body mass, including adipose tissue, caused the average systolic and diastolic blood pressure to decrease significantly by 3.6% and 5.4%, respectively.

Similar favorable changes in BMI, waist circumference, WHtR, and body fat percentage were obtained by Strączek et al. [28], even though a longer (lasting one year) dietary intervention based on nutrition education was provided to Polish children and/or their parents. The authors also noted a significant improvement in children’s eating habits, mainly due to a reduction in the consumption of unhealthy food products like sweets and sugar-sweetened beverages, sweetened dairy products, and sugary breakfast cereals. Other dietary interventions conducted among children and adolescents succeeded in achieving a reduction in pediatric obesity, measured with various indicators [25,29], although not all were successful [30]. Some of them were combined with increased physical activity. In the study by van Middelkoop et al. [29] conducted among children aged 6–12 years with excessive body weight, 12 weeks of intervention, including increased physical activity, showed a decrease in BMI z-scores by an average of 4% and a decrease in waist circumference by 0.2 cm. After 52 weeks of the intervention, only BMI z-scores decreased by another 4%, while waist circumference increased by 2.9 cm. On the contrary, in a study conducted among children aged 7–13 years, a 6-week intervention with a slightly energy-reduced diet (average caloric value of 1400 kcal/day) had no significant effect on BMI, but half of the participants experienced a fat mass decrease of 5.4% on average [31].

Chen et al. [32] introduced a multidisciplinary lifestyle modification program in three stages, namely, knowledge building (the first 4 weeks), habit consolidation (5–12 weeks), and self-monitoring (13–20 weeks). Significant improvements were found in body weight, BMI, BMI z-score, and waist circumference, and the decreases in body weight, BMI, and BMI z-score were most prominent in the first two stages.

Actions to combat childhood obesity must be undertaken community-wide. The best approach considers the family context and creates an environment conducive to healthier behaviors [12,14]. Involvement of the whole family, especially mothers, in the nutrition education process seemed to be especially beneficial. Our study achieved improvements in dietary quality and a reduction in childhood obesity. However, a literature review that focused on dietary interventions for the prevention of childhood obesity indicated mixed effects in terms of changing the body mass index of children; some studies reported small body weight reductions, others reported reductions that were clinically irrelevant, and still others showed no effects at all [33]. It is worth noting that the majority of studies were school-based interventions, with some addressing the whole community, and some interventions were implemented in the food sector and through mass media, which directly or indirectly could help to manage childhood obesity.

In our own study, a reduction in blood pressure was observed in the children after the 9-week intervention. Scientific studies have repeatedly proven the relationship between obesity and hypertension, and this has also been proven in children. In 42.9% of children with overweight and in 75% of children with obesity, elevated blood pressure was detected in the study by Wolske et al. [34]. Taking into account the process of weight reduction, the results of the meta-analysis showed that reducing body weight and BMI in children with obesity was associated with a significant change in systolic blood pressure [35]. An analysis of 42 studies conducted among 3807 children with an average age of 12.2 years showed that a reduction in SBP by 1 mm Hg was significantly associated with a decrease in BMI by 0.16 kg/m^2^. In relation to our results and others, blood pressure measurement in overweight children at developmental age can be treated as an important prognostic element for blood pressure values and other metabolic disorders in later life.

We also observed an improvement in dietary habits after 9 weeks of intervention. Similar results were obtained in a study conducted among 303 children aged 4 to 10 years [36]. The latter study demonstrated a positive association between changes in the Diet Quality Index (DQI) and the 10-week nutritional education program conducted by a dietitian. Improved adherence to recommendations for total fat, saturated fat, and sodium was noted. However, no change was observed for the consumption of vegetables and fruit, carbohydrates, and calcium. Considering the relationship between DQI and anthropometric parameters, a study conducted among 1700 children aged 9–10 years showed that a better-quality diet was significantly associated with a lower body weight, lower BMI, lower waist circumference, and lower fat mass [22]. This indicates that a better-balanced diet is associated with improved nutritional status and may therefore contribute to the prevention or treatment of childhood obesity. Excessive body weight is associated with an imbalance between energy expenditure and food intake, but the role of overall diet quality is increasingly being emphasized and appears to have a greater impact on health outcomes [37]. Growing evidence supports a shift from a nutrient-based approach to a pattern-based (and food-based) approach. Paying more attention to “diet quality” rather than “nutrient quantity” is a more reliable indicator in managing healthy eating [38].

Although the scale of the problem of excessive body weight among the pediatric population has been steadily increasing over the last several decades and various interventions have been undertaken, it is still not possible to combat it effectively. Analysis of the determinants of excessive body weight shows their complexity and multidimensionality, which in turn affect the appropriate course of treatment. The process of body weight reduction itself is also complex and long-term, and its success is determined by many factors. It is also important to consider barriers to implementing dietary changes in children, which may impact the achievement of goals and the completion of a weight-loss program. As Magalhães et al. [39] pointed out, children most often perceive family and community-related factors, such as self-regulation/emotional regulation and food availability at home, as barriers to adopting a healthy diet. Children reported difficulty in controlling the urge to eat certain foods, primarily unhealthy ones, due to their “presence” (smell and sight) in the environment, for example, while walking down the street. Furthermore, children mentioned that the habit of eating tasty foods (e.g., fast food) makes it difficult to incorporate healthier, less salty options into their diet. This suggests that children may lack the self-efficacy and self-regulation skills needed to focus on the goal of maintaining a healthy diet. These studies also highlighted that children’s social environments, such as easy access to local grocery stores and fast-food restaurants, can pose barriers to healthy eating. Furthermore, study participants expressed difficulty in avoiding restaurants they frequently visit or to which they are invited by friends and classmates. Summarizing the above findings and our own research, motivational factors, along with teaching children self-regulation (“When I’m hungry, I choose a salad over sweets”), are crucial for adopting a healthy diet. It also seems important to identify public health and health policy measures, including those limiting the opening of stores near schools that primarily offer unhealthy foods high in sugar, salt, and fat.

The results of our own study showed that the applied dietary intervention brought some positive effects; however, in order to achieve a significant improvement in the normalization of anthropometric parameters and identify more factors influencing the process of weight loss, it seems necessary to indicate its strengths and weaknesses.

A strong point of our study was the inclusion of individual dietary counselling that was adjusted to the needs of both the child and the parent/guardian. In addition, various dietary guidelines that were provided by the dietitian resulted in improved eating behaviours and thus in the achievement of reductions in body weight, BMI z-score, waist circumference, fat mas, and blood pressure.

On the other hand, the weaknesses of our study include the quite short duration of the dietary intervention and the too low frequency of follow-up visits. Moreover, non-compliance with nutritional guidelines by all children participating in the study and poor support from their parents/guardians were observed, which indicates the need to use additional motivation and incentive strategies. A longer period of study would allow us to determine whether these have an effect on the diet and intervention, because it is normal to experience some weight loss in the first stage of a diet. However, the body gets used to it and there is no significant weight loss after a few weeks. Furthermore, including biochemical measurements (e.g., lipid profile, fasting plasma glucose, and inflammation markers) would provide deeper insight into changes in nutritional and health status. This could explain inconsistent changes in body weight and increase motivation to adhere to the diet and/or continue with the program. Finally, our study lacked a control group, making it impossible to determine the percentage of observed changes attributable solely to the dietary intervention. Therefore, a control group without ongoing dietary support should be included in the future studies.

In addition, due to the interdisciplinary nature of the problem of pediatric obesity and the process of body weight reduction, it seems important to conduct multicomponent behavioural interventions that combine features addressing diet, physical activity, sedentary behaviours, sleep hygiene, and behavioural components in children and their families. Tailoring interventions to various subgroups based on age, gender, and culture might be needed. For example, with young children the activities might be largely parent-focussed, while for adolescents a greater degree of autonomy might be required [12,14]. Developing and strengthening intrinsic motivation and self-efficacy may bring additional advantages [40].

## 5. Conclusions

A 9-week dietary intervention based on an appropriate, high-quality diet in line with the Healthy Eating Pyramid guidelines, with individual counselling and constant monitoring of implemented changes, an individual approach, support from a dietitian, and the involvement of parents or guardians, can be an effective tool for starting to improve the diets and health of overweight children.

Undoubtedly, long-term, multifaceted programs with stronger behavioral components, greater parental involvement, and a broader set of health-promoting activities, such as limiting screen time and increasing physical activity and sleep duration, could be even more effective in creating lasting changes.

## Figures and Tables

**Figure 1 metabolites-15-00621-f001:**
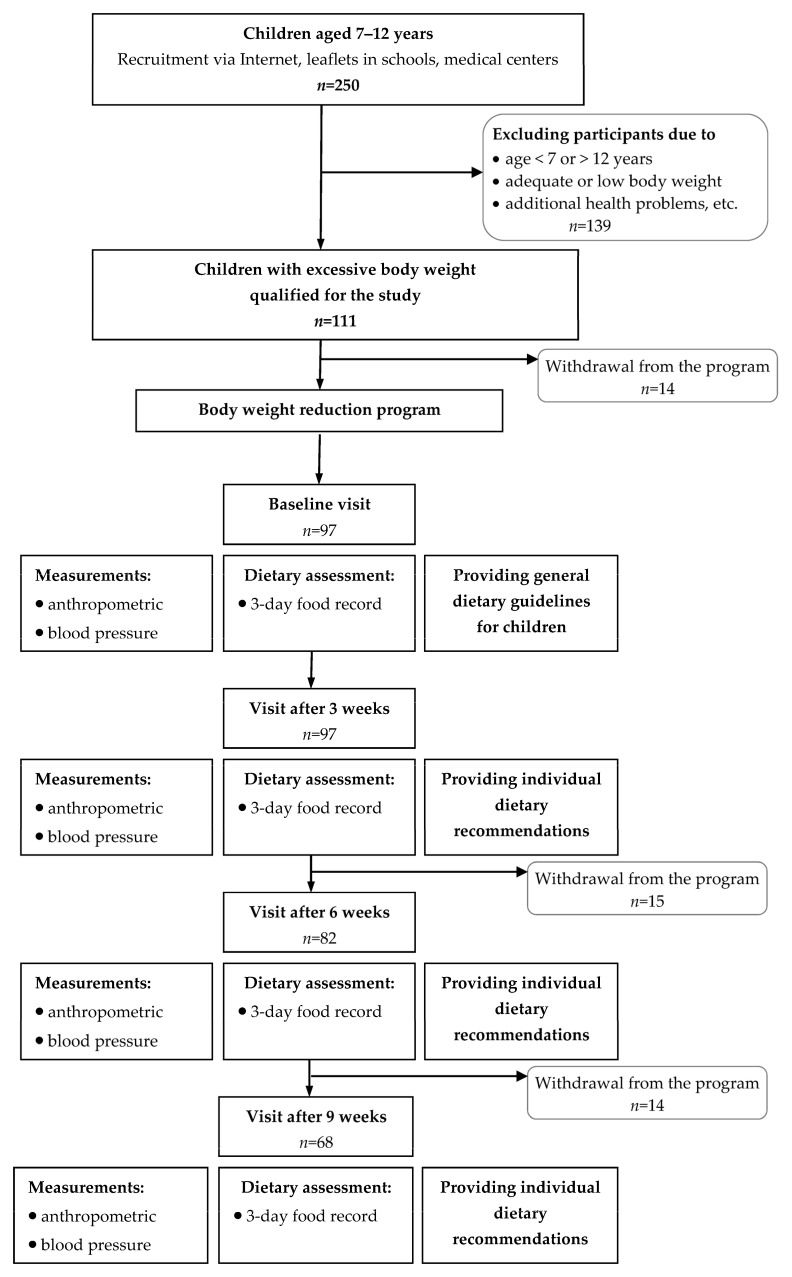
Flow chart of sample collection and study design.

**Table 1 metabolites-15-00621-t001:** Component score intake of Diet Quality Index [22].

No.	Intake	Score		
		0	1	2
1	Total fat (% of energy/d)	<30	30–40	>40
2	Saturated fat (% of energy/d)	<10	10–13	>13
3	Cholesterol (mg/d)	<300	300–400	>400
4	Fruit and vegetables (servings/d)	≥5	3–4	0–2
5	Breads and cereals (servings/d)	≥6	4–5	0–3
6	Protein (% RDA)	<200	200–250	>250
7	Sodium (mg/d)	<2400	2400–3400	>3400
8	Calcium (% RDA)	>100	67–100	<67
Sum		0–16		

**Table 2 metabolites-15-00621-t002:** Socio-demographic characteristics and BMI status of population (as %).

Variable	Population			
	Total, *n* = 68	Girls, *n* = 38	Boys, *n* = 30	*p*-Value *
Children’s age: 7–9 years 10–12 years	41.2 58.8	48.1 51.9	33.3 66.7	NS
Living area: Town City	52.6 47.4	53.8 46.2	51.1 48.9	NS
Maternal age: 19–30 years 31–45 years	72.2 27.8	69.2 30.8	75.6 24.4	NS
Parental age: 19–30 years 31–45 years	51.5 48.5	55.8 44.2	46.7 53.3	NS
Maternal education: Vocational Secondary University	26.8 42.3 30.9	26.9 40.4 32.7	26.7 44.4 28.9	NS
Parental education: Vocational Secondary University	43.3 35.1 21.6	44.2 32.7 23.1	42.2 37.8 20.0	NS
Maternal BMI status: Normal weight Overweight Obesity	44.3 42.3 13.4	48.1 36.5 15.4	40.0 48.9 11.1	NS
Parental BMI status: Normal weight Overweight Obesity	45.4 36.1 18.5	50.0 26.9 23.1	40.0 46.7 13.3	NS

* Pearson chi-square test; NS—statistically nonsignificant (*p* > 0.05).

**Table 3 metabolites-15-00621-t003:** Changes in selected nutrient intakes due to dietary intervention in children.

Nutrients		Dietary Intervention		*p*-Value *	Change (%)
	Recommendations [18]	Baseline *n* = 68	9 Weeks *n* = 68		
Energy (kcal) ^1^	1550–2050	1899 ± 147	1612 ± 155		
		1936	1626	<0.001	↓ 16.1%
		1285–2294	1190–2085		
Protein % of total energy intake	10–20	18.9 ± 1.7	19.0 ± 2.0	NS	↑ 0.5%
		19.0	18.9		
		14.8–22.5	14.7–23.5		
Fat % of total energy intake	<30	28.0 ± 3.2	25.7 ± 2.9	<0.001	↓ 8.5%
		27.9	25.4		
		19.3–34.6	19.5–31.9		
Carbohydrates % of total energy intake	45–65	53.1 ± 3.5	55.4 ± 3.2	<0.001	↑ 4.3%
		53.3	55.4		
		46.3–63.7	49.0–61.7		
Saccharose (g)	<10% of total energy for total sugars	23.1 ± 7.7	20.0 ± 7.4	0.006	↓ 13.4%
		23.1	19.5		
		4.7–43.6	6.4–41.8		
Dietary fiber (g)	16 (7–9 years) 19 (10–12 years)	15.0 ± 3.0	16.7 ± 3.2	0.001	↑ 11.3%
		15.1	16.3		
		7.1–22.5	10.9–26.3		
Calcium (mg)	800 (7–9 years) 1100 (10–12 years)	458.8 ± 128.3 454.7 158.9–863.5	507.7 ± 139.3 506.2 187.2–1027.3	0.021	↑ 10.7%
Iron (mg)	4 (7–9 years) 7–8 (10–12 years)	6.6 ± 1.1 6.5 4.5–10.2	7.9 ± 1.1 8.0 5.6–10.2	<0.001	↑ 19.7%
Vitamin C (mg)	40	52.7 ± 15.3 51.5 21.8–101.1	58.7 ± 23.0 55.9 27.6–135.4	<0.001	↑ 11.4%

^1^ Data are presented as means ± SDs, medians, and ranges; * Wilcoxon test; NS—statistically nonsignificant (*p* > 0.05).

**Table 4 metabolites-15-00621-t004:** Changes in DQI classification due to dietary intervention in children.

Population	Diet Quality	Dietary Intervention				*p*-Value *
		Baseline		9 Weeks		
		*n*	%	*n*	%	
Total *n* = 68	- Good	0	0.0	8	11.8	<0.001
	- Sufficient	60	88.2	60	88.2	
	- Poor	8	11.8	0	0.0	
Girls *n* = 38	- Good	0	0.0	4	10.5	0.014
	- Sufficient	36	94.7	34	89.5	
	- Poor	2	5.3	0	0.0	
Boys *n* = 30	- Good	0	0.0	4	13.3	0.002
	- Sufficient	24	80.0	26	86.7	
	- Poor	6	20.0	0	0.0	
***p*-Value ****		NS		NS		

* Pearson chi-square test; ** McNemar’s test; NS—statistically nonsignificant (*p* > 0.05).

**Table 5 metabolites-15-00621-t005:** The dietary intervention and anthropometric measurements/indexes in children.

Group	Dietary Intervention				*p*-Value *	Change (%)
	Baseline	3 Weeks	6 Weeks	9 Weeks		
BMI z-score						
Total ^1^ *n* = 68	2.80 ± 1.1 ^a^	2.59 ± 1.1 ^b^	2.56 ± 1.0 ^c^	2.50 ± 1.1 ^d^	0.001	↓ 10.7%
	2.51	2.41	2.44	2.44		
	0.7–5.6	0.5–5.6	0.4–5.3	0.3–5.1		
Girls *n* = 38	2.80 ± 1.1 ^a^	2.60 ± 1.1 ^b^	2.51 ± 0.9 ^b^	2.40 ± 1.0 ^c^	0.001	↓ 14.3%
	2.52	2.40	2.44	2.37		
	1.0–5.6	0.8–5.6	0.8–5.3	0.8–5.1		
Boys *n* = 30	2.72 ± 1.1 ^a^	2.57 ± 1.1 ^b^	2.62 ± 1.1 ^b^	2.64 ± 1.2 ^c^	0.001	↓ 3.0%
	2.52	2.41	2.42	2.63		
	0.7–5.6	0.5–5.4	0.4–5.1	0.3–5.1		
***p*-Value ****	NS	NS	NS	NS		
**LMS (%)**						
Total *n* = 68	151.9 ± 18.9 ^a^	148.8 ± 18.4 ^b^	148.2 ± 17.8 ^c^	146.7 ± 18.4 ^d^	0.001	↓ 3.4%
	149.1	146.2	145.7	144.9		
	116–197	113–197	112–192	111–189		
Girls *n* = 38	150.3 ± 19.5 ^a^	146.9 ± 18.6 ^b^	145.0 ± 16.5 ^b^	143.0 ± 17.1 ^c^	0.001	↓ 4.9%
	146.5	142.9	141.4	141.2		
	120–197	117–197	117–192	117–189		
Boys *n* = 30	153.6 ± 18.2 ^a^	151.8 ± 18.1 ^b^	151.8 ± 18.8 ^b^	151.3 ± 19.2 ^c^	0.001	↓ 1.5%
	151.4	151.3	149.9	152.3		
	116–194	113–191	112–186	111–186		
***p*-Value ****	NS	NS	NS	NS		
**WC (cm)**						
Total *n* = 68	87.7 ± 10.0 ^a^	86.7 ± 10.2 ^b^	86.2 ± 9.6 ^c^	85.2 ± 9.8 ^d^	0.001	↓ 2.9%
	86.0	86.0	85.0	83.5		
	66–112	66–112	68–107	63–107		
Girls *n* = 38	86.2 ± 10.3 ^a^	85.3 ± 10.5 ^b^	84.3 ± 9.2 ^c^	83.3 ± 9.9 ^d^	0.001	↓ 3.4%
	84.0	83.0	82.0	82.0		
	66–112	66–112	68–107	63–107		
Boys *n* = 30	89.5 ± 9.4 ^a^	88.5 ± 9.6 ^b^	88.3 ± 9.8 ^c^	87.5 ± 9.3 ^d^	0.001	↓ 2.2%
	89.0	90.0	88.5	87.5		
	72–110	71–106	70–105	69–103		
***p*-Value ****	NS	NS	NS	NS		
**WHtR**						
Total *n* = 68	0.53 ± 0.04 ^a^	0.52 ± 0.04 ^b^	0.52 ± 0.04 ^c^	0.51 ± 0.05 ^d^	0.001	↓ 3.7%
	0.52	0.52	0.51	0.50		
	0.42–0.66	0.43–0.65	0.43–0.64	0.43–0.64		
Girls *n* = 38	0.52 ± 0.04 ^a^	0.51 ± 0.04 ^b^	0.51 ± 0.04 ^c^	0.50 ± 0.04 ^d^	0.001	↓ 3.8%
	0.52	0.51	0.50	0.50		
	0.42–0.63	0.43–0.61	0.43–0.60	0.43–0.60		
Boys *n* = 30	0.54 ± 0.04 ^a^	0.53 ± 0.04 ^b^	0.53 ± 0.05 ^c^	0.52 ± 0.05 ^d^	0.001	↓ 3.7%
	0.53	0.53	0.52	0.52		
	0.47–0.66	0.47–0.65	0.45–0.65	0.45–0.64		
***p*-Value ****	NS	NS	NS	NS		

^1^ Data are presented as means ± SDs, medians, and ranges; NS—statistically nonsignificant (*p* > 0.05); LMS—Cole’s index; WC—waist circumference; WHtR waist-to-height ratio; * Friedman test, ^a–d^ values not sharing the same superscript in a row are significantly different (*p* < 0.05); ** Mann–Whitney U test.

**Table 6 metabolites-15-00621-t006:** The dietary intervention and fat mass measurements in children.

Group	Dietary Intervention				*p*-Value *	Change (%)
	Baseline	3 Weeks	6 Weeks	9 Weeks		
FM (kg)						
Total ^1^ *n* = 68	19.7 ± 7.5 ^a^	18.6 ± 6.9 ^b^	18.5 ± 6.5 ^c^	18.2 ± 6.6 ^c^	0.001	↓ 7.6%
	18.4	17.6	17.2	16.8		
	7–43	7–37	7–35	7–36		
Girls *n* = 38	18.7 ± 7.9 ^a^	17.6 ± 7.1 ^b^	17.0 ± 6.3 ^c^	17.0 ± 6.7 ^c^	0.001	↓ 9.1%
	16.8	16.0	15.8	15.6		
	7–43	7–37	7–35	7–36		
Boys *n* = 30	20.9 ± 6.8 ^a^	19.8 ± 6.5 ^b^	20.1 ± 6.4 ^b^	19.7 ± 6.2 ^b^	0.001	↓ 5.7%
	20.0	19.0	20.6	19.7		
	10–39	10–33	9–32	10–31		
***p*-Value ****	NS	NS	0.032	NS		
**FMI (kg/m^2^)**						
Total *n* = 68	9.0 ± 2.8 ^a^	8.5 ± 2.6 ^b^	8.4 ± 2.5 ^c^	8.4 ± 2.6 ^c^	0.001	↓ 6.7%
	8.5	7.9	7.9	7.8		
	5–17	5–16	4–15	5–15		
Girls *n* = 38	8.7 ± 2.9 ^a^	8.2 ± 2.6 ^b^	7.9 ± 2.4 ^b^	7.9 ± 2.5 ^b^	0.001	↓ 9.2%
	8.1	7.5	7.4	7.2		
	5–17	5–16	5–15	5–15		
Boys *n* = 30	9.4 ± 2.6 ^a^	8.9 ± 2.5 ^b^	8.9 ± 2.6 ^bc^	8.9 ± 2.6 ^c^	0.001	↓ 5.3%
	9.1	8.7	8.9	8.7		
	5–16	5–14	4–14	5–14		
***p*-Value ****	NS	NS	NS	NS		

^1^ Data are presented as means ± SDs, medians, and ranges; NS—statistically nonsignificant (*p* > 0.05); FM—fat mass; FMI—fat mass index; * Friedman test, ^a–c^ values not sharing the same superscript in a row are significantly different (*p* < 0.05); ** Mann–Whitney U test.

**Table 7 metabolites-15-00621-t007:** The dietary intervention and blood pressure measurements in children.

Group	Dietary Intervention				*p*-Value *	Change (%)
	Baseline	3 Weeks	6 Weeks	9 Weeks		
SBP (mm/Hg)						
Total ^1^ *n* = 68	109.2 ± 11.4 ^a^	107.9 ± 9.2 ^a^	104.7 ± 10.7 ^b^	105.3 ± 7.9 ^ab^	0.001	↓ 3.6%
	109.0	107.3	105.0	104.3		
	69–140	77–127	73–129	87–125		
Girls *n* = 38	108.8 ± 10.7	107.5 ± 7.7	104.8 ± 10.8	104.7 ± 8.4	NS	↓ 3.2%
	107.2	106.3	104.7	103.7		
	86–133	94–126	73–129	87–125		
Boys *n* = 30	109.7 ± 12.2 ^a^	107.0 ± 10.8 ^a^	104.5 ± 10.8 ^b^	106.2 ± 7.2 ^ab^	0.028	↓ 3.2%
	111.0	109.0	106.2	106.3		
	69–140	77–127	73–121	87–119		
***p*-Value ****	NS	NS	NS	NS		
**DBP (mm/Hg)**						
Total *n* = 68	73.8 ± 9.9 ^a^	71.3 ± 7.4 ^a^	69.9 ± 7.9 ^b^	69.8 ± 7.6 ^b^	0.002	↓ 5.4%
	73.7	70.3	69.0	69.7		
	43–114	57–91	56–88	48–84		
Girls *n* = 38	72.8 ± 8.4	71.4 ± 7.7	70.1 ± 8.4	69.1 ± 8.7	NS	↓ 5.1%
	72.0	71.0	69.2	69.0		
	53–89	58–91	56–88	48–84		
Boys *n* = 30	74.9 ± 11.5 ^a^	71.1 ± 7.1 ^a^	69.6 ± 7.4 ^b^	70.6 ± 5.8 ^b^	0.010	↓ 5.7%
	74.7	70.0	69.0	71.5		
	43–114	57–89	58–88	61–82		
***p*-Value ****	NS	NS	NS	NS		

^1^ Data are presented as means ± SDs, medians, and ranges; NS—statistically nonsignificant (*p* > 0.05); SBP—systolic blood pressure; DBP—diastolic blood pressure; * Friedman test, ^a,b^ values not sharing the same superscript in a row are significantly different (*p* < 0.05); ** Mann–Whitney U test.

## Data Availability

The raw data supporting the conclusions of this article will be made available by the authors upon request due to privacy.

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
