# Peer review of "The Effect of 9-Week Dietary Intervention on Anthropometric Parameters and Blood Pressure in Children with Excessive Body Weight"

_metabolites, 2025, doi:10.3390/metabo15090621_

Round 1
Reviewer 1 Report
Comments and Suggestions for Authors
In this manuscript, children with over weight were participated in 9 week program and intervention to evaluate its effects mainly on the blood pressure of the participants. It is valuable research as this issue is the main and growing concern particularly in developed and industrial countries. However, there are some comments could improve the scientific value of this manuscript.
It could be better to have more long period of study time to see the clear effects of this diet and intervention, as it is normal to have some weight loss in the first stage of the diet, but the body get use to and there would be not significant weight loss after some weeks. It would be better to have LDL/HDL or at least total cholesterol and blood sugar checked to have good discussion and their effects on blood pressure as well. As blood pressure is affected by the mentioned factors as well.
As there is a significant number of participants which were not continued to be in this study, please discuss the reason of this issue. Because, possibly there are some limitation or difficulties in this intervention or diet that possibly is not acceptable by the participants and consequently by the general public.
Please delete the repetition in introduction and shorten the texts.
There are two Table 3, please recheck it.
Reviewer 2 Report
Comments and Suggestions for Authors
Study by Karolina Gajda et al. “The effect of 9-week dietary intervention on anthropometric parameters and blood pressure in children with excessive body weight”
The authors performed and assessed the effect of the 9-week dietary intervention on anthropometric measurements and blood pressure among children aged 7-12 years with excessive body weight. The authors in the study evaluated a 9-week dietary intervention aimed at improving eating habits and health outcomes in Polish children with excess body weight. A total of 68 children (38 girls, 30 boys) participated, receiving personalized dietary guidance based on national nutrition guidelines, with regular support from a dietitian and active involvement from parents. The study measured diet, body composition, and blood pressure which were taken at the start and at 3, 6, and 9 weeks into the program. By the end of the intervention, participants showed significant improvements- 10.7% reduction in BMI z-score, 7.6% decrease in fat mass, and 4.0% reduction in body weight. Diastolic blood pressure also decreased notably, alongside observable improvements in dietary behaviors.
The authors suggest that a structured, individualized nutrition program supported by professionals and families can be an effective strategy for managing childhood overweight and obesity.
The author in introduction outlines the growing global concern of childhood obesity, highlighting its prevalence, associated health risks, and the urgent need for early intervention. It draws on a foundation of recent data from the WHO, COSI, and HBSC studies, emphasizing the biological, environmental, and behavioral contributors to obesity. The rationale for dietary interventions is clearly established, supported by recent EASO and EFAD recommendations on personalized, dietitian-led approaches.
The introduction is overly long and could benefit from more concise phrasing. Some sections, especially around environmental and dietary contributors, seem repetitive. While global and European data are mentioned, the introduction would be stronger with a clearer focus on the specific context or statistics relevant to Polish children, given the local scope of the intervention. The introduction would benefit from a better articulation of how this 9-week intervention adds to or differs from previous studies, especially since it acknowledges that past results are inconclusive.
The intervention consisted of personalized dietary plans based on Polish nutritional guidelines, focusing on gradual weight reduction (0.5 kg/week), healthy eating habits, and family involvement. Participants received four sessions of individualized dietitian counseling, written materials, and ongoing support. No additional physical activity was introduced to isolate dietary effects. Data was collected at baseline and at 3, 6, and 9 weeks. This included 3-day food records, anthropometric measurements (BMI, waist/hip circumference, WHtR), body composition (via BIA), and blood pressure. Dietary quality was assessed using the Diet Quality Index (DQI). Statistical analyses were conducted using SPSS, employing appropriate tests for non-parametric data. A significance level of p ≤ 0.05 was applied.
There are few issues in the methodology sections. Methodology may need more detailed analysis of dropout reasons and whether completers differed from non-completers in key baseline characteristics. The study lacks a control or comparison group. A future randomized controlled design would strengthen the evidence of effectiveness. While follow-ups and counseling were mentioned, the methods for objectively tracking participants adherence to dietary plans between visits are vague. The Diet Quality Index (DQI) scoring system is introduced, but the rationale behind the thresholds and point assignment is not properly explained.
In result section parental obesity was more frequently reported among fathers, particularly in families of girls, although no significant sex-based differences were observed in socio-demographic characteristics. Significant improvements were noted in energy and nutrient intake—energy intake dropped by 16%, fat by 8%, and saccharose by 13%. Fiber and micronutrient intake improved by over 10%. Girls showed greater relative improvements than boys across most metrics. However, weight loss averaged 0.2 kg per week lower than the targeted 0.5 kg/week. Both systolic and diastolic blood pressure levels decreased by 3.6% and 5.4% respectively with the most significant reductions observed after 6 weeks.
Without a control group, it’s difficult to determine how much of the observed changes can be attributed solely to dietary intervention. The 9-week period is relatively short. It remains unclear whether the positive effects would be sustained over time, especially without ongoing support.
In discussion section study explored how a structured 9-week dietary intervention, guided by dietitians and involving parental support, influenced weight management and health outcomes in children aged 7–12 with excessive body weight.
The study needs whether the improvements were sustained over time or how behaviors evolved after the program ended. The average weight loss of 0.2 kg/week was below the targeted 0.5 kg/week which is still beneficial, this may point to either overly goals or areas for improving adherence and engagement. The study noted inconsistent parental support yet parents are key influences in children's diets. More structured or mandatory parental involvement might enhance outcomes. Without blood tests or metabolic markers, it's hard to understand the full health impact of the intervention (e.g. cholesterol, insulin sensitivity, inflammation markers).The intervention focused heavily on diet, but did not address physical activity, screen time, sleep habits, or motivation strategies which are key elements in sustained behavior change, especially for children
The study presents a practical model for improving health in children with excessive weight through tailored, family-supported dietary changes. However, longer-term, multi-faceted programs, with stronger behavioral components, greater parental involvement, and a broader set of health measures, would likely be even more effective in creating lasting change.
Reviewer 3 Report
Comments and Suggestions for Authors
The paper analyzes in a scientific manner the progress of research on child nutrition on reducing the risk of developing obesity by analyzing the content of nutrients with a main role in the harmonious growth and development of children, and the summarized strategies are truly useful for the development of specific food policies to increase the level of nutritional education.
I suggest that this paper can be considered accepted after minor revisions, which mainly include the supplementation of data of interest specific to the nutritional study.
Although the subject of this analysis is relevant and aligns with the current interest in the role of nutritional education in the formation of healthy eating behaviors, the manuscript in its current form presents several limitations:
1. I recommend that the authors introduce in Table 2 the total caloric content of the diet as recommended values ​​for the age group as well as the caloric content of the diet as a result of the recommendations of the dieticians involved. This aspect would lead both to a better understanding of the effects of nutritional education based on dietary recommendations on improving health and reducing the risk of obesity in children and as a guideline for parents and children in correcting risky eating behaviors among children and adolescents.
In this sense, readers can understand what the recommended amount of macronutrients means, specifically for reducing the degree of obesity (the % amount of macronutrients reported to the total energy available per day).
Moreover, the authors state that: ,,During the dietary intervention, significant and positive changes were demonstrated in terms of energy intake and selected nutrients, except for proteins (line 307-308),, but energy intake is not presented to highlight the degree of improvement in the diet and implicitly in eating behavior.
2. In the conclusions, I recommend that the authors use the degree of appreciation of the improvement in diet and health as a percentage value and not the expression "many children improved their diet.....", thus the study will have a great scientific impact by highlighting what % of the total improved their health.
3. I recommend rewriting the conclusions so as to highlight the effects of such a study on risky eating behaviors on childhood obesity, specifying more specifically the role of nutritional education.
